# SALON: Simplified Sensing System for Activity of Daily Living in Ordinary Home

**DOI:** 10.3390/s20174895

**Published:** 2020-08-29

**Authors:** Tomokazu Matsui, Kosei Onishi, Shinya Misaki, Manato Fujimoto, Hirohiko Suwa, Keiichi Yasumoto

**Affiliations:** 1Nara Institute of Science and Technology, Ikoma, Nara 630-0192, Japan; onishi.kosei.of5@is.naist.jp (K.O.); misaki.shinya.mq9@is.naist.jp (S.M.); manato@is.naist.jp (M.F.); h-suwa@is.naist.jp (H.S.); yasumoto@is.naist.jp (K.Y.); 2RIKEN, Center for Advanced Intelligence Project AIP, Chuo-ku, Tokyo 103-0027, Japan

**Keywords:** energy harvesting sensor, daily activity recognition, machine learning, simple installation sensing system

## Abstract

As aging populations continue to grow, primarily in developed countries, there are increasing demands for the system that monitors the activities of elderly people while continuing to allow them to pursue their individual, healthy, and independent lifestyles. Therefore, it is required to develop the activity of daily living (ADL) sensing systems that are based on high-performance sensors and information technologies. However, most of the systems that have been proposed to date have only been investigated and/or evaluated in experimental environments. When considering the spread of such systems to typical homes inhabited by elderly people, it is clear that such sensing systems will need to meet the following five requirements: (1) be inexpensive; (2) provide robustness; (3) protect privacy; (4) be maintenance-free; and, (5) work with a simple user interface. In this paper, we propose a novel senior-friendly ADL sensing system that can fulfill these requirements. More specifically, we achieve an easy collection of ADL data from elderly people while using a proposed system that consists of a small number of inexpensive energy harvesting sensors and simple annotation buttons, without the need for privacy-invasive cameras or microphones. In order to evaluate the practicality of our proposed system, we installed it in ten typical homes with elderly residents and collected the ADL data over a two-month period. We then visualized the collected data and performed activity recognition using a long short-term memory (LSTM) model. From the collected results, we confirmed that our proposed system, which is inexpensive and non-invasive, can correctly collect resident ADL data and could recognize activities from the collected data with a high recall rate of 72.3% on average. This result shows a high potential of our proposed system for application to services for elderly people.

## 1. Introduction

Recently, various approaches to sustainable development goals (SDGs) [1], which are considered blueprints for achieving a more sustainable future, and Industry 4.0 [2], which aims at promoting technological innovations, have been pursued around the world. In Japan, such efforts have been conducted in Society 5.0 [3], which aims at facilitating both economic developments and finding solutions to social problems through a system that fuses cyber and physical spaces. To realize Society 5.0, cyber-physical systems (CPSs) need to be more thoroughlydisseminated in various fields and locations.

Such systems are designed to provide high-quality services and help resolve social problems by sensing/collecting data in real-space, and then storing the collected data in cyber-space by utilizing the Internet of Things (IoT), as well as information and communication technology (ICT). In particular, issues that are related to aging populations, which are becoming increasingly serious worldwide, must be resolved because aging rates in suburban towns and residential areas are progressing faster than national averages [4]. As a consequence, there are urgent needs for effective measures to (1) monitor and care for elderly people and (2) promote the extension of their healthy lifestyles.

To resolve the above-mentioned issues, a number of studies have examined the use of activity recognition technologies as part of efforts to provide monitoring services for elderly people and improve their lifestyles. For example, Aran et al. [5] proposed an anomaly detection system for elderly people, while Alcal et al. [6] proposed a monitoring system that is based on home appliances usage. Additionally, De et al. [7] proposed a micro-activity recognition system based on wearable devices and smartphones.

However, many of the existing activity recognition systems based on CPSs are expensive or require specific equipment. Additionally, since they do not take into account the maintenance and usability of long-term sensing, there is a problem in spreading to ordinary homes. Therefore, existing systems have not reached the point of collecting and analyzing data in an in-the-wild environment.

To collect in-the-wild data, it is necessary to construct CPSs that can be installed in ordinary homes. As part of such efforts, a variety of activity of daily living (ADL) systems have been proposed for monitoring elderly people in smart homes. These include data collection systems (hereafter referred to as ADL sensing systems) [8,9], camera-based ADL recognition systems [10], and fall detection systems [11]. However, smart homes that require specific equipment [8,9,12,13,14] also have the following three issues: (1) they are generally expensive; (2) installing the systems in typical homes with different floor plans is difficult; and, (3) the use of cameras and/or microphones has the potential to infringe on resident privacy. One typical smart home kit that was developed on the premise that it would be utilized in ordinary homes is the Smart Home in a Box (SHiB) [15] system that was produced by the Center for Advanced Studies in Adaptive Systems (CASAS). This system has gained popularity, because it is inexpensive, does not violate user privacy, and can be used in residences with different floor plans.

However, the use of the CASAS SHiB kits results in two additional issues: (4) it requires regular maintenance, because all of the sensors are battery-powered; and, (5) its use requires the completion of a complex annotation process. As a result, the installation and use of this system impose a burden that is too high for many elderly people. It should also be noted that many elderly people are particularly reluctant to use systems that require wearable devices, systems with low operational reliability, and systems with complex interfaces that are difficult to understand [16]. Hence, there is an ongoing need for ADL sensing systems that can operate over long periods of time without the need for battery replacements and can operate without requiring complex interactions with the home residents.

In this paper, we propose a novel senior-friendly ADL sensing system that can solve the five issues that are mentioned above. To realize our proposed system, it was necessary for the system to fulfill the following five requirements (which cope with the above issues (1)–(5)): (i) it should be inexpensive, (ii) it should not be affected by the installation location or floor plan of the residence, (iii) it should generally protect user privacy, (iv) it should operate maintenance-free for long periods of time, and (v) it should have a simple user interface that enables elderly people to easily put annotations on activities.

Req. (i) is achieved by using only relatively inexpensive motion and environmental sensors. Req. (ii) is achieved by using only small wireless sensors instead of special furniture or home appliances. Req. (iii) is achieved by collecting only binary data such as human-detection sensor activations and environmental information such as temperature and humidity, without requiring cameras or microphones. Req. (iv) is achieved by adopting environmentally powered (energy harvesting) sensors that do not require batteries as system components. Req. (v) is achieved by using simple pushbuttons that are powered by energy harvesting and do not require annotation software.

In order to confirm that our proposed system fulfills all the requirements mentioned above, we installed it in ten ordinary homes occupied by elderly people and conducted a two-month experiment. Additionally, to confirm the applicability of the collected data to services for elderly people, we performed activity recognition processing for the five basic activities of “bathing”, “cooking”, “eating”, “going out”, and “sleeping,” using deep learning models. The main contributions of this paper are summarized as follows:First, we constructed a sensing system that fulfills Reqs. (i)–(v) by assuming that the system would be used by typical elderly people. Because our proposed system consists of energy harvesting sensors, long-term driven environmental sensors, and simple annotation pushbuttons, it is senior-friendly and does not violate user privacy. The use of maintenance-free sensors allows for long-term data collection (Reqs. (i), (iii), and (iv)). Additionally, because the proposed system is not affected by residence floor plans, it can be easily installed in a wide variety of different homes (Req. (ii)). Additionally, the resident’s burden for annotation can be reduced by using simple pushbuttons unlike existing works such as CASAS SHiB or Kasteren’s system that use complicated software or wearable mic devices for annotation (Req. (v)). Through installation and operation in ten typical homes occupied by elderly residents, we confirmed that our proposed system fulfills Reqs. (i)–(v). Furthermore, we confirmed that our proposed system could correctly collect ADL data by visualizing the relationships between the ADL and sensor data that were collected during the experiment.Second, we analyzed in-the-wild ADL data collected by our proposed system in typical homes and confirmed that it could recognize activities at a recall rate of approximately 72%.

The rest of this paper is organized, as follows. Section 2 reviews the existing work on activity recognition methods in home environments and ADL sensing systems that are related to our study, while Section 3 presents our proposed sensing system, and Section 4 describes the data collection experiment using the proposed system. Section 5 describes the analysis method of the collected data and its results. Section 6 describes the discussion and limitation of this paper. Finally, we conclude this paper in Section 7.

## 2. Related Work

In this section, we review existing work related to our study. First, we examine the requirements that must be met before an ADL sensing system can be disseminated into typical living environments for elderly people, and then describe the living space areas that are covered by existing activity recognition and ADL sensing systems. Finally, we clarify the positioning of this work by comparing our proposed system with those existing sensing systems.

### 2.1. ADL Sensing System Requirements for Homes of Typical Elderly People

In this subsection, we provide the requirements that must be met in order to realize an ADL sensing system that can be installed in general home environments for elderly people. A number of sensing systems aimed at recognizing the activities of elderly people in their living spaces have already been proposed, as mentioned in Section 1. However, most of those have only been investigated in experimental environments. Additionally, those systems are relatively expensive, have limited installation locations, and often require specialized equipment.

Other systems utilizing cameras and other privacy-invasive sensors have also been proposed, but their use in ordinary residences is problematic. Additionally, although the CASAS SHiB [15] kit was developed for ordinary home use, that system is unsuitable for elderly people in many cases, because it regularly needs maintenance such as changing battery. Furthermore, activity annotations, which are an essential part of ADL data collection, are often difficult for elderly persons because the processes typically require the residents to operate complex software applications.

With these points in mind, we determined that the following requirements would need to be fulfilled before an ADL sensing system could be widely disseminated into the residences of elderly persons. **Req.** **(i):**It should be inexpensive.**Req.** **(ii):**It should not be affected by installation locations or residence floor plans.**Req.** **(iii):**It should generally protect user privacy.**Req.** **(iv):**It should operate maintenance-free for long periods of time.**Req.** **(v):**It should have a simple user interface that can be easily annotated by elderly people.

In the following sections, as part of efforts to clarify the positioning of this study, we provide a detailed investigation of existing activity recognition and ADL sensing systems that are designed for use in living spaces, after which we discuss whether these existing systems fulfill the above-mentioned five requirements.

### 2.2. Activity Recognition Systems Used in Living Spaces

As mentioned above, numerous systems that are based on activity recognition technologies that estimate resident activities and their contexts have already been proposed [17,18,19,20]. Basically, these systems can be roughly divided into two categories according to whether or not they require the use of wearable devices. Currently, there is a wide variety of systems that incorporate the use of wearable devices, smartphones, or both [7,21,22,23,24,25]. These include a system that combines smartphones with environmental sensors for recognizing the contexts of resident activities [22], and a system that uses unique wearable devices to provide for healthcare monitoring services for elderly people [24].

However, systems that include the use of wearable devices are often unsuitable for elderly people due to their complexity and the burden of constantly wearing them. Therefore, in order to realize a senior-friendly ADL sensing system, it is first necessary to develop a system that does not include wearable devices. With that point in mind, there are a number of studies that have proposed systems that do not include wearable devices, but, instead, attempt to recognize the activities of elderly people via non-contact devices.

For example, there are many systems in which radio waves are used. These include activity and gesture recognition systems using Wi-Fi [26,27,28] or radio frequency identification (RFID) systems [29,30,31,32] and a healthcare reporting generation support system that is designed for nursing homes that uses Bluetooth low energy (BLE) components [33,34]. However, because these systems are directly affected by radio wave reflections, they need to be reconfigured every time the environment or device location changes. As a result, they are not suitable for operations other than experimental environments and, thus, do not fulfill Req. (ii) mentioned above.

As for ADL sensing systems for elderly people, an anomaly detection system that is designed to detect signs of dementia [35], an anomaly detection system for monitoring the actions of elderly people [5], and a camera-based fall detection system [11] have been proposed. While all of these are specialized for anomaly detection, systems that require the use of cameras are especially problematic because they may violate user privacy and, thus, do not fulfill Req. (iii). Even when a system uses cameras, it can protect privacy by extracting features, such as bounding boxes or skeletons from the captured images and discarding the images. It should be examined whether the target user feels discomfort or not despite the fact that the use of cameras can be expected to improve the accuracy of recognition. There have also been some studies involving wireless sensor networks aimed at the objective of providing improved healthcare for elderly people [36]. However, because these proposed systems are based on experimental/specific environments rather than ADL sensing systems in typical home environments, they do not fulfill Req. (ii).

### 2.3. ADL Sensing System for Restricted Environment

As methods of obtaining ADL data, there are systems that use lifeline data, such as electricity and water consumption levels, as well as systems in which various sensors have been embedded to create smart homes. In the example involving utility usage rates, recognizing resident lifestyle activities from water flow information [37,38], combining it with appliance usage information and sensor data [39,40,41,42,43], have been proposed to estimate ADL data. These systems also need equipment and additional sensors that must be prepositioned in the home. Because many of the systems that are discussed above use a large number of sensors, or a variety of expensive sensors that can only collect data only in suitable experimental environments, a more natural sensing system based on ADL data that can be installed in the actual homes of elderly people is needed in order to promote healthy life extension in those residents. Additionally, to create a systematic data set in a smart home, there are some systems, which are ARAS [9], Domus [13], and Sweet Home [14]. These studies do not fulfill Reqs. (i) and (ii).

### 2.4. ADL Sensing System in General Environments

An important perspective in the development of an ADL sensing system for elderly people residing in ordinary homes is the consideration of privacy infringements. Because home ADL data are intensely private by nature, many studies have proposed methods to ensure confidentiality while maintaining security [44,45,46,47,48,49,50]. Nevertheless, regardless of the resident’s generation, the collection of images and sounds may be considered invasive, so systems that require such data do not fulfill Req. (iii). Therefore, to the greatest degree possible, it is necessary that the sensing systems installed in living environments only collect information that does not invade the privacy of the residents, and that measures be taken in order to reduce the risk of information leakage and the sense of invasiveness.

One current option is SHiB, which is a kit that is designed to be used in ordinary homes. As of this writing, more than 60 datasets have been collected and published using SHiB kits [51]. The target residences were ordinary homes with elderly residents, families living with pets, and houses with multiple residents. Each SHiB kit consists of cables for electronic equipment, relays, temperature sensors, magnetic door sensors, motion sensors, batteries, and adhesive tape. Because the sensors can be installed at any location, the system is not affected by the residence floor plan. In addition, the sensing system is relatively inexpensive to install, since the kits are available for just a few thousand dollars.

The evaluation results of elderly people who have installed SHiB kits in their homes by themselves [52] have generally been favorable. This indicates that it is possible to introduce such kits into ordinary homes that are occupied by elderly people. However, all of the SHiB sensors are battery-powered and require periodic maintenance. In addition, because the system requires approximately 40 sensors, installation is time-consuming, and many residents report feeling unsettled by the invasive level of observations after its installation. Furthermore, in order to annotate their activities, residents need to first define them by themselves using software that runs on a personal computer (PC). However, since it has been pointed out that elderly people tend to feel burdened by the need to perform system maintenance, and because it is often difficult for elderly people to use PCs or operate applications with complex interfaces, it appears that SHiB kits do not fulfill Reqs. (iv) and (v).

SPHERE [53] provides an ADL sensing system for elderly people living in ordinary homes in the UK. The purpose of the system is to provide healthcare services for the elderly, and there are several examples of its use [54]. It generally consists of low-priced wearable devices, RGB-D cameras, PIR, and ambient sensors. The wearable device is quite small, and it is designed to reduce the obtrusiveness of wearing it. A simple mobile app is developed to annotate activity, and it also reduces resident’s reluctance. In contrast, the system needs to be tuned for each floor plan, and it requires regular maintenance several times a year. Although the system fulfills Req. (i), and partially fulfills Reqs. (iii)–(v), it does not fulfill Req. (ii).

### 2.5. Position of Our Study

Table 1 shows a comparison of existing ADL sensing systems that cover areas related to our work. From this table, it can be seen that, although each item has been partially considered in existing systems, no sensing system has been built that meets all five of the above-mentioned requirements. Therefore, in this study, we aimed to construct a senior-friendly ADL sensing system that is suitable for typical home environments that can easily collect data while fulfilling those requirements.

## 3. Proposed System

In this section, we propose a senior-friendly ADL sensing system that can fulfill Reqs. (i)–(v). We will begin by describing the design of our proposed system, after which we provide a detailed description of the system configuration.

### 3.1. System Design

In this section, we describe our system design in order to demonstrate how it meets Reqs. (i) through (v). To meet Reqs. (i) and (ii), we designed a relatively inexpensive system that was equipped with compact sensors and a data server to ensure it could be easily installed in ordinary homes. To meet Req. (iii), we restricted the sensors that were used in our system to those that collect binary data, such as motion reactions and door opening/closing information, or data that are not directly related to user privacy, such as temperature, humidity, and noise levels. To fulfill Req. (iv), we adopted sensors that operate via energy harvesting. Finally, to fulfill Req. (v), we adopted simple pushbutton devices for purposes, such as activity annotation. Based on the above, our proposed system consists of the following five components:Motion sensors (energy harvesting)Ambient sensors that collect temperature, humidity, illuminance, pressure, and noise informationMagnetic door sensors (energy harvesting)Pushbuttons for annotation (energy harvesting)Data server (small PC)

### 3.2. System Configuration

Figure 1 shows an overview of the proposed system for each home. The system consists of motion sensors, ambient sensors, door sensors, annotation pushbuttons, and a data collecting server. Each of these components will be described below.

#### 3.2.1. Motion Sensors

Passive infrared (PIR) motion sensors perform their detection function by receiving the infrared radiation emitted from living bodies. Therefore, it is important to note that the sensor may operate incorrectly if it is subjected to body heat from pets or temperature changes that result from direct sunlight. To produce a sensor that can be operated without maintenance over long periods of time, we combined the EKMB1101112 (https://www3.panasonic.biz/ac/e/search_num/index.jsp?c=detail&part_no=EKMB1101112) PIR sensor made by Panasonic with the energy harvesting module of the STM431J made by Rohm (STM431J is a temperature sensor with the energy harvesting module, but we do not use the temperature sensor: https://www.enocean.com/en/products/enocean_modules_928mhz/stm-431j/data-sheet-pdf/). This sensor enables sensing and data transmission to the server using only environmentally generated power. The sensor is similar in size to a typical USB flash drive, and it is quite light (4.5 g) because the cover is made of resin. Most medium-sized rooms can be covered with a single emplaced sensor because the sensor has a 94 degree horizontal, 82 degree vertical, and 5 m deep detection range.

#### 3.2.2. Ambient Sensors

For use as an ambient sensor, we adopted the 2JCIR-BL (https://www.components.omron.com/product-detail?partNumber=2JCIE-BL) made by Omron, because it can collect a variety of data, is small in size, and is capable of operating for long periods of time. According to the datasheet that was provided by the manufacturer, this sensor is about the same size as two motion sensors and weighs approximately 16 g. When integrated into our proposed system, the sensor collects temperature, humidity, illuminance, pressure, and noise information for transmission to the data server via BLE. Because the data transmission range of a BLE device is 10 m, the sensor can maintain a connection to the data server when installed in most ordinary homes. The sensor also has a built-in memory function that allows it to continue collecting data when temporary communications problems occur, and a battery life of six months when a 5-min. sampling rate is set.

#### 3.2.3. Door Sensor

To acquire the door opening/closing data, we adopted the STM250J (https://www.switch-science.com/catalog/2549/) magnetic sensor made by Rohm. This device, which is driven by energy harvesting, is composed of two parts: the sensor body and the magnet. The sensor acquires door opening/closing information when the sensor body that is attached to the door detects the magnet attached to the door frame. The sensor body then sends the acquired opening/closing information to the data server.

#### 3.2.4. Annotation Pushbutton

To annotate the resident’s ADL, we adopted the PTM210J (https://www.zaikostore.com/zaikostore/en/stockDetail?stockID=ST37823807) pushbutton data transmission sensor made by Rohm. The sensor performs energy harvesting using an electromagnetic induction module that generates power using the force exerted when a button is pushed and then uses that energy to send a data packet to the server via EnOcean. Each sensor button is circular in shape and it has two switches. Each switch is marked with a sticker to indicate the activity start and end states. Residents can annotate the start and end of the activity by pushing the appropriate switch.

#### 3.2.5. Data Server

For data server use, we adopted the NUC (https://www.intel.com/content/www/us/en/products/boards-kits/nuc.html) small form factor PC made by Intel. A BLE dongle and EnOcean are used for communication with the server. The data from ambient sensors are received via BLE, while motion sensor, door sensor, and annotation button data are received via EnOcean. The received data are stored in Influx DB [55], which is a time-series database running in the server.

To confirm whether the sensing system is working during operation, we use ngrok [56], which is a software development and deployment tool, in order to create secure tunnel requests from a public URL to our local system. Here, a long-term evolution (LTE)-compatible PIX-MT100 (http://www.pixela.co.jp/products/network/pix_mt100/spec.html) Universal Serial Bus (USB) dongle made by Pixela is connected to the data server, which is then connected online via a mobile network.

## 4. Data Collection Experiment

In order to confirm whether the proposed system that is described in Section 3 is capable of collecting ADL data while fulfilling Reqs. (i)–(iii), we conducted experiments in ordinary homes occupied by elderly people. Here, it should be noted that confirming Reqs. (iv) and (v) required long-term experiments in those same homes. In this section, we begin by describing the experimental conditions and the sensor installation conditions. Subsequently, we describe the characteristics of each of the experimental homes and discuss the factors that affected our analysis. Finally, we visualize the data obtained from our experiments and confirmed that our proposed system was capable of collecting ADL data.

### 4.1. Experimental Objective

The objective of our experiment was to confirm whether our proposed system could be deployed as expected in ordinary homes, and whether it could collect data that can be used to monitor elderly people and recognize their activities. The Ethics Review Board of the Nara Institute of Science and Technology approved this experiment (Reception Number: 2018-I-26).

### 4.2. Experimental Process

The experiment was started after explaining the purpose of the research to the participants and obtaining their consent. Each sensor was installed in accordance with the floor plan of the home. After installation, checks were made in order to ensure proper communications between each sensor and the data server. The participants of the experiment used the annotation buttons that were distributed to them to label their ADL. A single annotation button was used for each activity if the residence had a single occupant. When there were two residents in the same home, they shared a single annotation button for each activity. For example, when two residents started eating at the same time, they pushed the start annotation button twice. Likewise, the end button was pushed twice at the end of the activity. If a resident forgot to push the appropriate button at the appropriate time, he or she was instructed to make a written entry on the daily questionnaire. The experimental duration was two months for each home. However, if the residents were outside of their homes for more than a single day, they were requested to extend the experimental period so that it would last a full two months. After the two-month experimental period, we collected the system from the participating homes and concluded the experiment.

### 4.3. Experimental Target

The experimental targets were ten homes that were occupied by elderly people in the vicinity of Ikoma City, Nara Prefecture. Of the ten targets, three were single-occupant dwellings, and seven were two-occupant homes. All of the participants were in their 60 s to 80 s and they were selected from public volunteers. The purpose, method, and significance of the experiment, along with the voluntary nature of consent, the possibility of consent revocation, and relevant privacy considerations, were explained in writing and orally to all participants prior to the experiment, and written consent in the form of signed consent forms was obtained from each. For each system setup, motion sensors (6 to 10), ambient sensors (7 to 10), and door sensors (1 or 2) were installed according to the floor plan of the home. Here, it should be noted that door sensors could sometimes not be installed due to the shape and materials of the door. However, generally speaking, the installation process, including participant briefings, sensor installation, and data server setup, was all completed within one hour.

Table 2 shows the characteristics of each home in which the sensor was installed. Based on the results of our experiment, we confirmed that our system could be installed in ordinary homes, regardless of the floor plan, thus satisfying Req. (ii). Additionally, we found that the occupants of each home had different lifestyles. For example, the occupants of five of the target homes had almost no visitors, while the remaining five homes sometimes/often received guests. As a result, there were cases in which guests who were not the targets of our sensing experiments performed actions, such as bathing or eating. In such cases, the system collects sensor information as ADL data from these external guests. Additionally, since two of the homes had a cat in residence, the sensors may have sometimes reacted incorrectly due to their actions.

### 4.4. Installation of the Proposed System

The installation of our proposed system was carried out by actually visiting each of the participating homes. Because these homes all had different floor plans, we defined simple installation conditions for each. Annotation buttons were placed at the location where each activity was expected to be performed. The installation conditions for each sensor are shown below.**Motion Sensor**: We prepared a maximum of ten for each home and installed no more than one in each room, where they were arranged to provide the maximum coverage, in order to reduce the number of motion sensors. However, in some cases, some rooms did not have sensors. The contact surface of the sensors was protected by masking tape, and each device was fixed to a wall with double-sided tape (Figure 2). This process made it easy to install and remove the sensors without damaging the wall surface. Each sensor was placed at a height of 1 m from the floor in order to be clear of obstacles, such as furniture and curtains. This installation procedure also ensured that the sensor would react to people while avoiding pets and home appliances that could cause false activations. **Ambient Sensor**: Ambient sensors were emplaced in optional locations at a height of 1 m from the floor in the same manner (and in most cases adjacently to) the motion sensors (Figure 2). The sampling interval was set at 3 min. (1/180 Hz) in consideration of the trade-off between the experimental period and the battery life. In our preliminary experiment, we had confirmed that the sensor could operate continuously at this sampling rate over a period of three months. **Door Sensor**: Door sensors were installed at the entrance and bathroom doors using the same procedure that was described for the motion sensor. The main body of the sensor was installed on the door, and the magnet was installed on the door frame (Figure 3). **Annotation Pushbutton**: In this experiment, five activities (“bathing,” “cooking,” “eating”, “going out”, and “sleeping”) were selected as sensing targets. Therefore, five annotation button sets were placed in each home. The installation locations were chosen to make it as easy as possible for the participants to push the proper annotation button at the correct times. For instance, the annotation button for “cooking” was placed near the kitchen, and the annotation button for “sleeping” was placed near the bed. The participants were instructed to push the start/end switches at the start/end times of the target activities. In addition to using the annotation buttons, the residents were asked to fill in daily questionnaires (Figure 4) at the end of the day to annotate their ADL activities. In the questionnaire, the residents were asked to confirm whether they performed each activity and whether they had remembered to push the annotation buttons appropriately. **Data Server**: To maintain high strength communication levels, the data server was placed in an unobstructed location, such as under a sofa in the center room of the home (Figure 5).

### 4.5. Data Collection Results

In this subsection, we describe the sensing results that were obtained via the proposed system. Figure 6 and Figure 7 show the ADL data collected from a single-resident home, as well as the number of motion sensor reactions and noise levels in the main rooms. These collected ADL data indicate that the participant woke up at 06:00 a.m. and started preparing breakfast. After that, the participant went out from 09:00 a.m. and remained out until the evening. He then prepared dinner, took a bath, and went to bed at around 23:00 p.m. Focusing on the number of motion sensor reactions in the kitchen, we confirmed that the activation frequency during “cooking” is higher than the other sensor reactions. Similarly, we can confirm higher levels of reactions in the living room during meals and in the bedroom during “sleeping”. In addition, the frequency of reactions in the kitchen was higher before and after “bathing”. This is due to the fact that the kitchen and the bathroom are located next to each other. Therefore, we consider it proven that the sensor could detect the movements of the participants.

Focusing on the noise level, we found that noise increased in the kitchen after meals, primarily because the sensor detected washing dish activity. Furthermore, because the noise level of the kitchen next to the bathroom increased during “bathing”, we confirmed that the noise sensor could collect bathing information that was not detected by the motion sensors. From these results, we concluded that our proposed system could collect sensor data that can provide an adequate overview of resident ADL data and would, thus, be useful for anomaly detection and/or activity recognition.

## 5. Analysis of Activity Recognition

In this section, we used a deep learning model to examine activity recognition levels in order to ensure that the data collected by our proposed system can perform activity recognition with sufficient accuracy in ordinary environments. We will begin by describing the data imputation and preprocessing methods that are required for activity recognition. Then, we describe the algorithm of activity recognition and evaluation method using long short-term memory (LSTM). In a previous study [57], we reported the results of a preliminary analysis conducted on one home. In this analysis, the missing value imputation method is improved based on the results of that preliminary analysis, after which it is performed on all ten of the homes participating in our experiment.

### 5.1. Data Shaping

The sensors used in the proposed system have different sampling rates. Since the motion sensors, door sensors, and annotation buttons send packets when they react to stimuli or when they are activated, they do not have a set sampling rate. In contrast, the ambient sensor has a set sampling interval of 3 min. We begin by generating time windows to use these sensor data for the analysis. These time windows were empirically set to 10 s. The motion sensor and door sensor data were then stored based on the number of reactions during each time window.

As for activity labels, data were stored with “1” as the start and “0” the end. The last set value was maintained until the next time the annotation button was pushed. The ambient sensor data were stored in such a way that each measured value (i.e., temperature, humidity, illuminance, pressure, and noise) was retained at the last recorded level until the value changed. As an additional feature, the time window number from the start of the day to the end of the day was normalized from “0” to “1” and added to the dataset. This allows the model to recognize the time when each activity took place within the day.

### 5.2. Handling Missing Values

Although sensor data can be mechanically collected, activity label data are human-mediated, which means that data loss can occur if the participant forgets to push the appropriate button at the correct time. Activity label data correspond to ground truth data in the activity recognition process. However, if there are missing values, then the ground truth data will be incorrect. If that occurs, the model cannot learn properly, and the activity recognition accuracy level may be reduced. Therefore, the imputation of the missing information is essential. Because the activity label data are basically a set of start and end points, missing activity label data are categorized, as follows:(Type 1)Forgetting to push both start and end buttons.(Type 2)Forgetting to push the start button.(Type 3)Forgetting to push the end button.

Type 1 cannot be read out from the data at all. Therefore, it can only be imputed by a questionnaire report, which is a self-report made by the resident. Type 2 forgetting exists independently of the end data. Likewise, Type 3 forgetting exists independently of the start data. Hence, these data can be imputed by mechanical methods in addition to the daily questionnaire. We impute the missing values for each home and activity, as follows:(Step 1)Modify the activity data based on the questionnaire (corresponding to Types 1 to 3)(Step 2)Generate the start instance from independent end data using the average time of target activity (corresponding to Type 2)(Step 3)Generate the end instance from independent start data using the average time of target activity (corresponding to Type 3)

Figure 8a–c show the imputation examples of a correct instance, an instance when the resident forgets to push the start button (Type 2), and an instance when resident forgets to push the end button (Type 3). Because our proposed system distributes a fixed number (five) of annotation buttons to each household, in a home with two residents, there may be cases where two sequential start instances are followed by two sequential end instances. In such cases, the second end data value is adopted as the end time of the activity for both people. Additionally, going out activity occurs irregularly, so it is difficult to impute the missing value of that activity. Thus, the imputation process mentioned above is not performed in this study.

Figure 9 shows the total number of missing values for each activity in each participating household, as imputed by the flow described above. From the figure, it can be seen that even though there is some variation in missing values across the participating homes, the overall number of missing values for “sleeping” is low, while other activities have relatively high missing value rates. Table 3 shows the ratio of increase in activity duration between the data deleted over the threshold time and the data imputed by the proposed method. The activity duration available for learning is increased overall by utilizing imputation in the proposed method, specifically, 44% for bathing activity, 23% for cooking activity, 20% for eating activity, and 13% for sleeping activity, as shown in the table. From this result, we found that the ratio of increase in the activity duration on “sleeping” is smaller than that in the other activities. One likely reason is that “sleeping” is an activity of pause at the end of the day, so it is easy to link the activity and the annotation. In contrast, to more reduce the missing values, we found that it will be necessary to devise improved imputation methods for missing values and to develop a system that will reduce instances where the participants forget to push the correct buttons in a timely manner.

### 5.3. Algorithm for Activity Recognition

In this study, we constructed an ADL recognition model using LSTM in order to consider time-dependency in the collected data. Figure 10 shows the configuration of the deep learning model. The model consists of an input layer, a dense layer, a dropout layer, an LSTM layer, and an output layer. It is implemented by Keras, which is an open-source deep learning framework for Python with TensorFlow as the backend. As for the hyperparameters, the number of units on the dense and LSTM layers is 512, the dropout ratio of the dropout layer is 0.2, the lookback length of the LSTM layer is 100, the number of epochs is 10, and the batch size is 512.

As for activation functions, the activation function of the dense layer is a rectified linear unit (ReLU), the activation function of the LSTM layer is tanh, and the activation function of the output layer is sigmoid. Five binary classification models are generated for each activity, which is the number of activities. This procedure translates the multi-label problem in multi-resident homes into a binary classification problem.

Although the ADL data that were collected by our system are imbalanced, there are several methods that can be used to deal with such imbalances. These include learning with cost-sensitive methods, data augmentation methods, such as the Synthetic Minority Oversampling Technique (SMOTE) [58] model, and a threshold-moving method that changes the threshold value of the probability output from the model. In this paper, we adopted a cost-sensitive learning method for handling the imbalanced data. Specifically, we used the weight for cost-sensitive learning, as defined by the following:(1)$$w_{i}=\frac{N}{C_{n}\times F_{i}}$$
(2)$$W_{i}=\frac{1+w_{i}}{2}$$
where $W_{i}$ is the weight to be used for learning, $w_{i}$ is the weight of each classification class, *N* is the number of all samples, $C_{n}$ is the number of classes, and $F_{i}$ is the number of samples that belong to class *i*.

### 5.4. Evaluation Method

Next, we utilized a time-window based cross-validation method to evaluate the model while preserving the time-series of the data. From the starting point of the time-series data, one sequential data record is extracted for each number of the lookback length data (100 samples = 1000 s) set as a hyperparameter. A new series of data is then generated by sliding the extracted data every one sample (10 s). Of the generated time-series data, the first 80% is divided as train data, the next 10% is divided as validation data, and the last 10% is divided as test data. We used the test data to evaluate the binary classification model that was generated for each activity from the train and validation data, after which we calculated the precision, recall, and F measure.

### 5.5. Activity Recognition Results

Table 4 shows the precision, recall, F measure, standard deviation (SD) of the precision, and SD of the recall values for the “bathing”, “cooking”, “eating”, “going out”, and “sleeping” activities in each home. From the table, it can be read that, for all activities, there is a tendency to have high recall and low precision. Although the precision is relatively high for “going out” and “sleeping”, which have many data points, the precision is low for “bathing”, “cooking”, and “eating”, so we will discuss this result below.

The activity with the highest recall was “cooking”, and the activity with the lowest was “eating”. The reason is believed to be that cooking activities often take place in a specific room (the kitchen), while eating activities often take place in either the living room or dining room. For example, if a resident often eats while watching television, it is difficult to determine whether they are watching television or eating from the motion sensor information. In other words, when living and dining activities are combined, ADL activities other than “eating” can take place in the eating place, which may decrease recall. On the other hand, there are few types of activities that are commonly performed in kitchens other than “cooking”, so the model correctly recognized that the activities that were detected in the kitchen are most likely to be cooking-related, and we consider this fact to improve recall. For the same reason, focusing on the recall SD, we find that recall variation is large for “eating” and small for “cooking”. Regarding “going out”, even between single occupant homes and multi-resident homes/homes with pets, we find that precision and recall are not significantly different from other homes. Regarding “bathing”, we found large variations in the activity duration among the participating homes. Therefore, the evaluation metrics tend to be lower in homes with less duration. Regarding “sleeping”, we found that recognition accuracy tends to be higher, and the variation tends to be lower because the time for “sleeping” is clear, and the motion sensor reacts less often during sleep periods. From the sparse data that were collected by our proposed system, we confirmed that the activity recognition is possible at a high recall rate of 72.3%. However, improving the precision will be an important issue when considering its application to actual services.

## 6. Discussion and Limitation

In this section, we present the discussion and limitation. First, we explain discussion for result. Subsequently, we explain the limitation of our study. Finally, we explain the comparison with the related work.

### 6.1. Discussion

Although the proposed annotation interface is better than existing research that forces residents to use PC and/or headset for annotations, annotation task in our method is still burdensome to residents. We will consider introducing an active learning mechanism that identifies activities recognized with low confidence and then prompts the residents to annotate only such low confidence activities to further reduce the burden of annotation. At that time, it is possible to annotate activities in more natural way using an interface or a dialogue agent that can interact with the resident by voice or simple gestures.

There are several possible reasons for the low recognition accuracy. The first reason is the number of sensors. Existing smart home studies conducted experiments in rather special homes with various sensors, while our study targets ordinary homes with a small number of sensors easy to install. Because the information that is obtained through such small number of sensors is limited, we believe that the recognition accuracy is reduced. The second reason is the number of residents. Many previous studies targeted homes with a single person living alone. In this study, we believe that the accuracy was reduced, because the single occupant homes and couple homes are mixed in our dataset. The third reason could be due to lifestyle differences. Japanese homes are relatively small; hence, various different activities, like eating, watching TV, etc., are all done in the living room. Other possible reasons may include the age range of the participants. Improving the accuracy of recognition is our future work. As an idea of increasing accuracy, we will consider not only the generation of new features, such as the reaction order of the sensor and the context recognition approach, but also the situation recognition approach [59].

### 6.2. Limitation

In this paper, the sensing target is the relatively healthy elderly. In reality, many elderly people have cognitive and physical problems, and it is necessary to make assumptions about such problems [60]. The main target of the proposed system is the elderly, who are transitioning from a healthy state to one requiring long-term care. For elderly who completely require long-term care, an elderly person monitoring service, such as iCarer [61], is required. However, these existing services for the elderly who need nursing care can be strengthened by sensing ADLs.

Because sensing in a general home environment is prone to missing sensor data, the system has to consider dependability/reliability [62,63,64,65,66], including quality of network, device, and data, for effective learning of recognition models. To deal with missing data problem, a wireless network system with high reliability for communication and continuous operation is required. To detect system malfunctions, the system has to have functions, such as data visualization and anomaly detection. In addition, to deal with the case of leaking data caused by system vulnerabilities, the system should be designed to be able to carefully select the information for activity recognition [67]. We will cope with the issues and improve the system as future works.

### 6.3. Comparison with Related Work

In this section, we describe a comparison with related work according to concept, sensor installation, annotation, target activity, environment/data, and algorithm/result. Table 5 shows contents of the comparison. Because SPHERE [53] does not show the target activity and detailed analysis results, we do not list it in Table 5.

#### 6.3.1. Concept

Kasteren [8], ARAS [9], Placelab [12], and CASAS [15] aim to develop a kit that can make ordinary homes smarter. Each proposal has different concepts, such as the kit focusing on extendability, multi-residents, and improving the quality of annotation. Our proposed system is also aiming to develop the kit, additionally with the premise that it is useful for every person including elderly. In contrast, Sweet Home [14] and Domus [13] system aim to advance a more intelligent smart home system with multi-modal interaction, such as voice.

#### 6.3.2. Sensor Installation

Kasteren [8], Placelab [12], and CASAS [15] contain several binary sensors and/or ambient sensors. However, ARAS [9] collects more diverse sensor data using seven types of sensors at different places. Kasteren [8], ARAS [9], and CASAS [15] use approximately 20 sensors. Placelab [12] uses larger number of sensors. Our proposed system also deploys approximately 20 sensors. However, because some of the sensors drive with energy harvesting, the system does not require maintenance, which is a distinguishing attribute of our system apart from other works. In real-world households, the section boundaries of where a particular activity is taken are unclear. Additionally, since a large number of sensors are required for every addition of activity, sensor installation corresponding to specific activities is undesirable. We believe the proposed system is valuable in the sense that it provides more easy-to-deploy/manage sensors with energy harvesting than existing studies.

#### 6.3.3. Annotation

ARAS [9], Placelab [12], and CASAS [15] require software operations on PC or PDA for activity annotation. Kasteren [8] requires residents to wear bluetooth headset. Our proposed system adopts mechanical pushbuttons that can be installed anywhere and operated intuitively, even by the elderly.

#### 6.3.4. Target Activities

CASAS [15] collects approximately 10 types of annotated ADLs data. The number of annotated activities increases/decreases depending on the distributed data set. ARAS [9] system treats 26 types of activities, but actually, only six kinds of activities are handled in their analysis. Kasteren [8] refers to Katz ADL index [68] to define target activities and choose seven types of activities. However, they constrain that all activities are performed only within its given area; hence, it would be relatively easy to recognize the activities rather than other target activity settings. The Katz ADL index is an evaluation scale for the elderly. However, the test subject is a 26-year-old man, thus the evaluation targeting the elderly is not conducted. PlaceLab [12] considers a large amount of activities as well as ARAS [9], but the maximum number of target activities that could obtain high accuracy in machine learning was only eight. In this paper, we choose five basic ADLs, which are expected to contribute people’s lifestyle and easily applied to practical services such as healthcare.

#### 6.3.5. Environment/Data

We compare experiment fields that the data were collected in related works. Although CASAS [15] has tested in various residential environment, such as multi-resident, households with/without pets, the actual data they analyze is ADL data of 18 residents living alone in “smart apartment”, according to the paper. Therefore, the result of activity recognition is not examined with the data that were collected from ordinary homes. Regarding Kasteren [8], ARAS [9], and Placelab [12], they do not describe whether the data are collected from ordinary homes or not in their paper, but all of them indicate that the data is collected from one-story houses. Kasteren [8] illustrates a floor plan with sensor arrangement indicating that many sensors are installed mainly in bedroom, bathroom, toilet, and entrance. The sensor layout of ARAS [9] indicates that many kinds of sensors are also installed on each furniture. Likewise, regarding one of CASAS dataset [51], the sensor layout shows that approximately 10 sensors are totally installed in each room. In this paper, because most of target houses are not apartments but two-story structure, the floor plans are completely different among homes. In addition, the number of sensors installed in each room is up to three, and up to approximately 20 in total, as shown in Figure 11.

#### 6.3.6. Result of Activity Recognition

The recognition accuracy shown by Kasteren [8] is 94.5% with HMM and 95.6% with CRF, respectively. It shows considerably high accuracy, but the chosen activities are highly linked to the sensor locations. The recognition accuracy that is shown by ARAS [9] is 61.5% for House A and 76.2% for House B with HMM, respectively. In ARAS [9], the authors describe that it determined the sensor installation position based on the correspondence of each activity. Therefore, the system requires an appropriate decision of sensor location and reassignment of a larger number of sensors as increasing number of activities. The recognition accuracy that is shown by Placelab [12] is 64.3% for first participant and 50.8% for second participant, with NB (Naive Bayes) classifier, respectively. The recognition accuracy shown by Sweet home [14] is 85.3% with MLN (Markov Logic Network), which is relatively high accuracy. The machine learning approach using multimodal sensor data is helpful for activity recognition in home environments. These studies evaluate the results of activity recognition by accuracy, not precision, recall, or F measure. In general, in the case of dealing with ADL data, which is imbalanced data, evaluation by accuracy is not suitable, because difference in the number of samples between activity classes affects the evaluation result.

In CASAS [15], the result of activity recognition is evaluated with SVM (Support Vector Machine) and average of five-fold cross validation. The result shows that the F measure is 58.9% on average for all activities. However, paper [15] does not describe details of analyzed data and the data collection environment is a smart apartment that is different from our target environments (i.e., ordinary homes). Hence, it is difficult to make comparison by the recognition accuracy. In our proposed system, the result of activity recognition is evaluated by 8:1:1 holdout validation with LSTM, and the F measure is 40.7%. As mentioned above, our system has many factors that could lead to a decrease in accuracy (e.g., multiple residents, pets, guests, etc). Therefore, we regard that the F measure obtained is not so bad, because the results were obtained from the data collected under such factors.

## 7. Conclusions

In this paper, as part of efforts to construct a system for monitoring for elderly people and encouraging healthy and independent lifestyles, we proposed a monitoring system that is easy-to-install, inexpensive, and can operate independently and maintenance-free over long periods of time. We installed our proposed system into ten typical homes occupied by elderly people and carried out a two-month experiment during which we collected ADL data. It was confirmed that our system can be easily installed and it is capable of collecting data over long periods of time. To recognize resident ADL data, we constructed an LSTM model and analyzed the collected information. From the results obtained, we confirmed that the model could recognize ADL at a high recall rate of 72.3% on average for the five activities of “bathing”, “cooking”, “eating”, “going out”, and “sleeping”. However, we also found that there were missing values in the data used for learning due to problems, such as forgetting to push the correct annotation button at the stop/end of an activity. The data quality could be improved by designing a more advanced sensing system interface and finding ways to more naturally encourage residents to push the appropriate buttons at the proper times. As future work regarding annotation, it is valuable to develop a system that can be easily used by people with disabilities, such as printing braille on buttons for annotation and audio guidance. Accordingly, as an issue that is related to activity recognition methods, it will be necessary to study learning systems with higher accuracy levels.

## Figures and Tables

**Figure 1 sensors-20-04895-f001:**
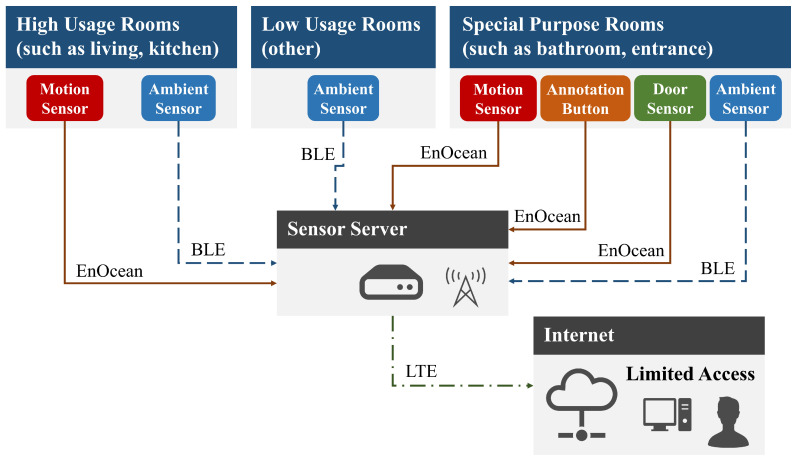
System overview.

**Figure 2 sensors-20-04895-f002:**
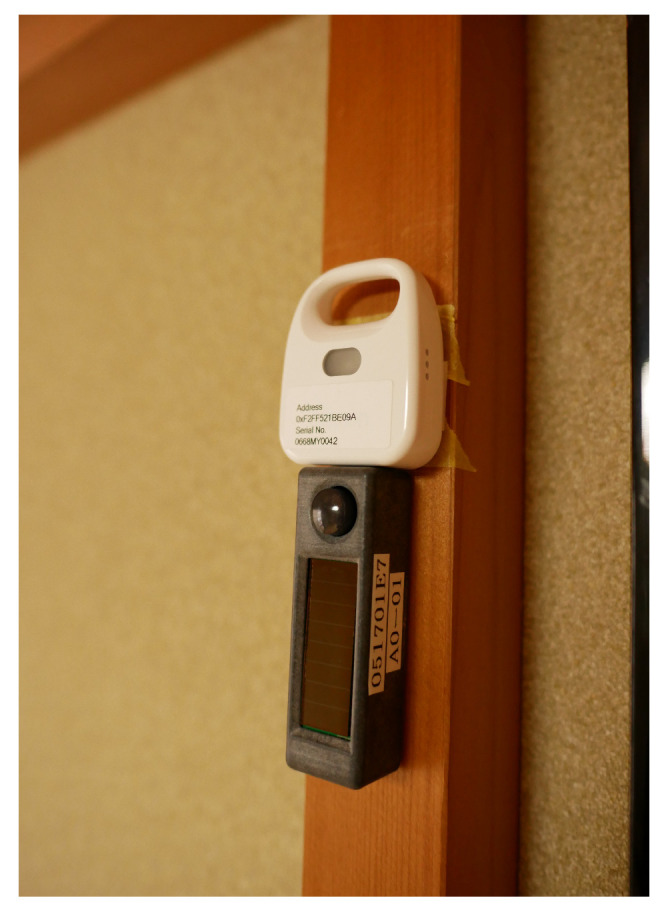
Ambient/motion sensor.

**Figure 3 sensors-20-04895-f003:**
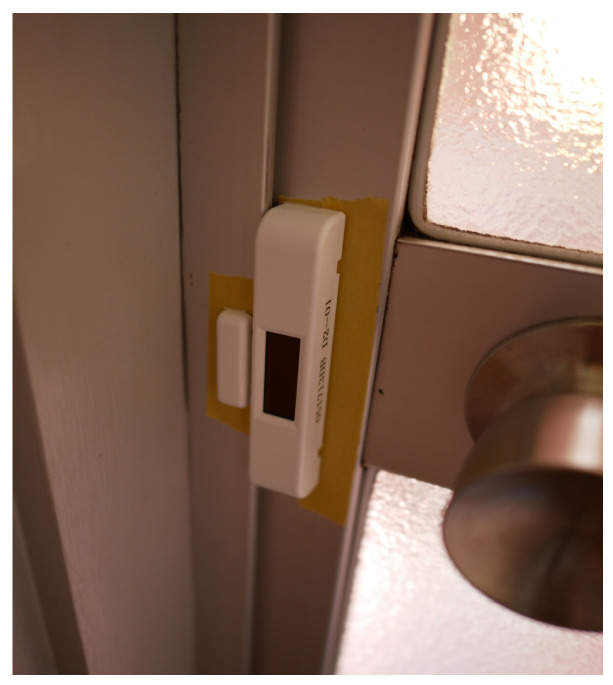
Door sensor.

**Figure 4 sensors-20-04895-f004:**
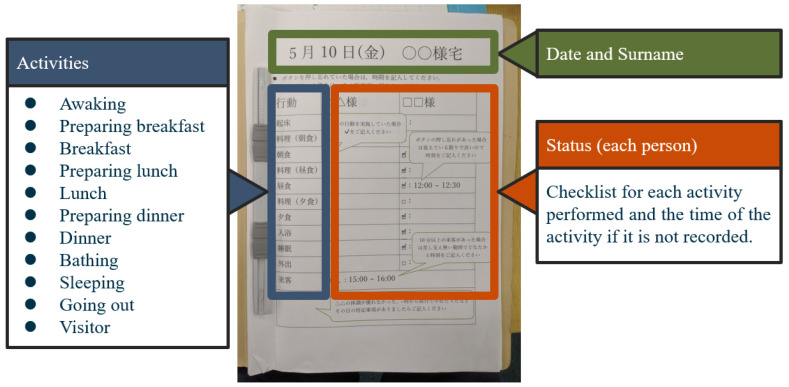
Daily questionnaire.

**Figure 5 sensors-20-04895-f005:**
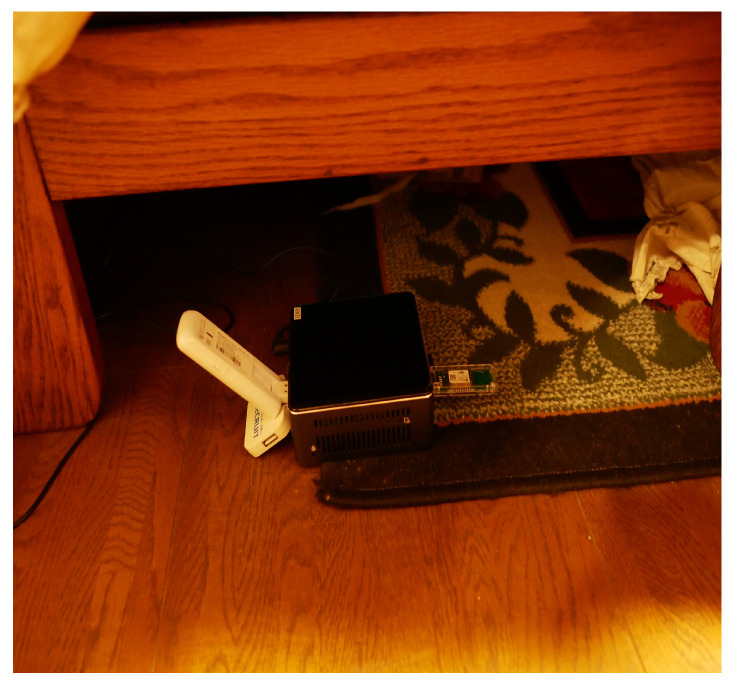
Data Server.

**Figure 6 sensors-20-04895-f006:**
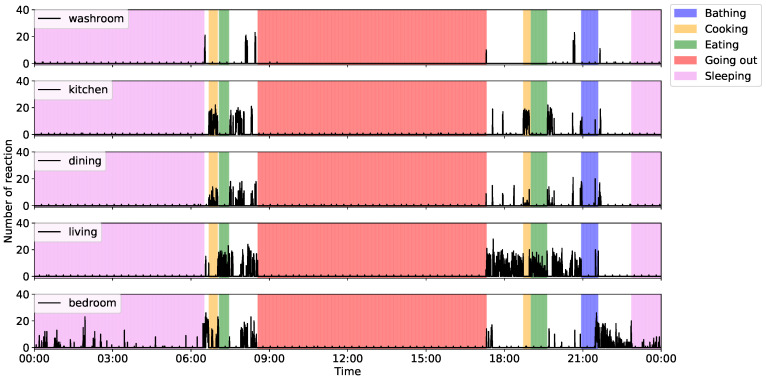
Example of collected activity of daily living (ADL) data (motion sensor).

**Figure 7 sensors-20-04895-f007:**
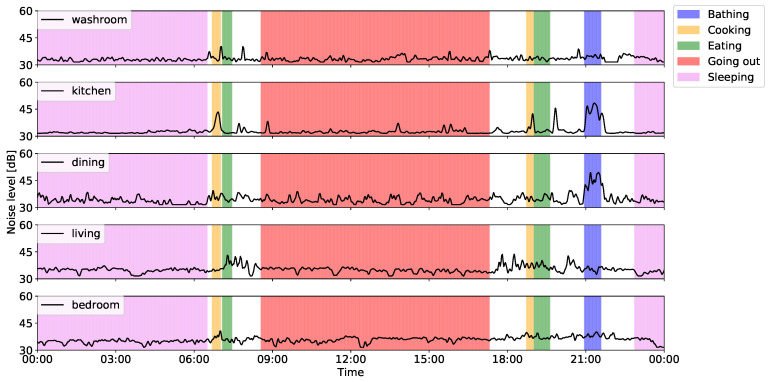
Example of collected ADL data (noise levels).

**Figure 8 sensors-20-04895-f008:**
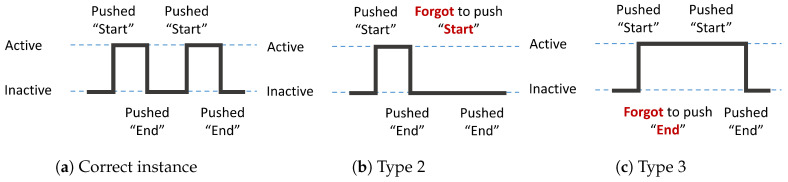
Imputation examples.

**Figure 9 sensors-20-04895-f009:**
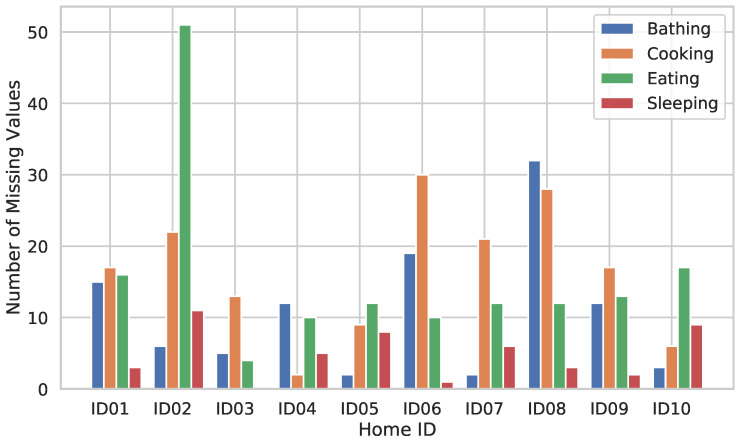
Number of missing values in each activity for each home.

**Figure 10 sensors-20-04895-f010:**
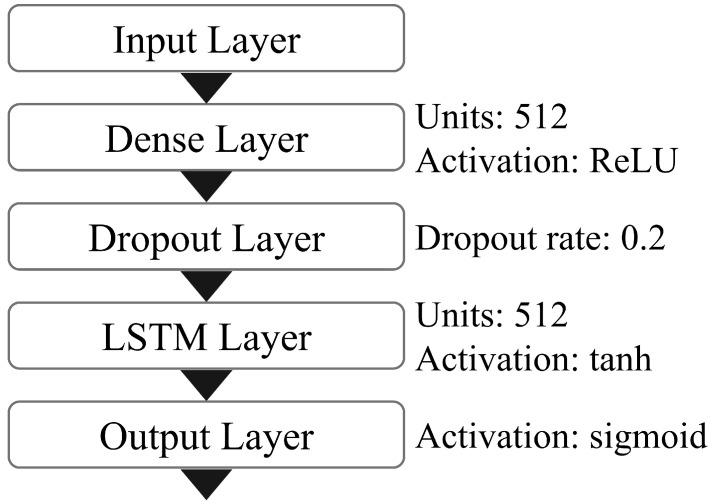
Long short-term memory (LSTM) network configuration.

**Figure 11 sensors-20-04895-f011:**
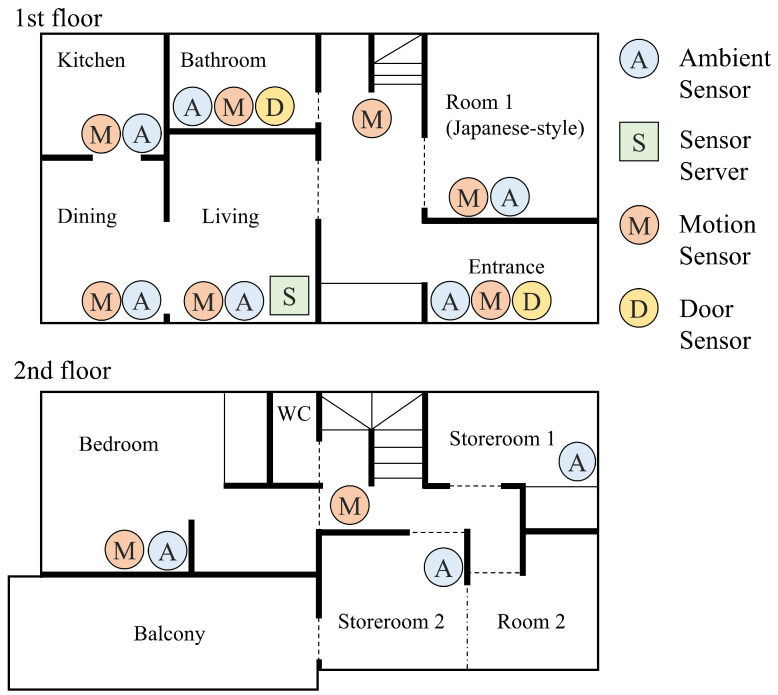
Floor plan of the target of proposed system (a selected house).

**Table 1 sensors-20-04895-t001:** Comparison with existing systems related to our work.

	Req. (i)	Req. (ii)	Req. (iii)	Req. (iv)	Req. (v)
Kasteren [8]	✓	✓	✓	△	n/a
ARAS [9]	✓	n/a	✓	△	n/a
Placelab [12]	✓	✓	✓	△	n/a
Domus [13]	n/a	n/a	n/a	n/a	n/a
Sweet Home [14]	n/a	n/a	n/a	n/a	n/a
CASAS SHiB [15]	✓	✓	✓	△	n/a
SPHERE [53]	✓	n/a	△	△	△
Our System	✓	✓	✓	✓	✓

✓: it fulfills the requirement. △: it partially fulfills the requirement. n/a: it does not fulfill the requirement.

**Table 2 sensors-20-04895-t002:** Specifications for each home.

	Number of Residents	Motion Sensors	Ambient Sensors	Door Sensors	Guests	Pet	Remarks
ID01	2	10	10	0	Often: grandchild, friend	No	-
ID02	2	8	7	0	Few or no	No	Sometimes bathing at health club.
ID03	2	10	10	1	Few or no	No	-
ID04	2	9	9	1	Few or no	Cat	One resident is woman in her 30 s.
ID05	1	10	10	2	Few or no	No	-
ID06	2	10	10	2	Often: grandchild	No	Grandchildren perform activities such as bathing or eating.
ID07	1	6	7	1	Often: child	No	Bad communication signal because of reinforced concrete.
ID08	2	10	10	0	Sometimes	No	Sometimes guest stays overnight.
ID09	2	10	10	2	Few or no	Cat	-
ID10	1	6	7	2	Rarely: caregiver	No	Regular caregiver visits.

**Table 3 sensors-20-04895-t003:** Ratio of increase in activity duration by imputation.

Activity	ID01	ID02	ID03	ID04	ID05	ID06	ID07	ID08	ID09	ID10	Ave.
Bathing	22.9%	22.6%	71.4%	11.8%	37.7%	32.5%	5.8%	207.8%	24.5%	10.1%	44.7%
Cooking	21.7%	48.2%	15.2%	18.3%	14.8%	33.1%	24.9%	12.5%	32.1%	10.1%	23.1%
Eating	5.0%	29.9%	26.1%	23.4%	29.1%	3.6%	19.7%	11.7%	24.7%	31.4%	20.5%
Sleeping	1.0%	11.4%	0.0%	8.3%	24.0%	19.4%	13.6%	10.1%	16.9%	26.9%	13.4%

**Table 4 sensors-20-04895-t004:** Metrics of activity recognition.

Activity	Precision	Recall	F Measure	Precision SD	Recall SD
Bathing	0.249	0.711	0.306	0.2628	0.2843
Cooking	0.212	0.850	0.310	0.1310	0.1547
Eating	0.198	0.549	0.254	0.1487	0.3053
Going out	0.351	0.720	0.424	0.2746	0.2828
Sleeping	0.824	0.787	0.740	0.2207	0.2486
Total avg.	0.367	0.723	0.407	0.2076	0.2551

**Table 5 sensors-20-04895-t005:** Overall comparison with related work.

	Concept	Sensor Installation	Annotation	Target Activity	Environment/Data	Algorithm/Result
Kasteren [8]	A smart home kit for high quality annotation.	• Digital sensors (Binary sensors) × 14. *Installation time: N/A*	*Bluetooth Headset*	Leave house, Toileting, Showering, Sleeping, Preparing breakfast, Preparing dinner, Preparing beverage.	Sensing term: 28 days Field: N/A (just a room) Number of participants: 1 Number of residents: 1	Algorithm: HMM, CRF *Result:* *Accuracy with HMM—94.5%* *Accuracy with CRF—95.6%*
ARAS [9]	A smart home kit for households with multiple residents.	• Force sensitive resistors • Pressure mats • Contact sensors • Proximity sensors • Sonar distance sensors • Photocells • Temperature sensors • IR receivers *Installation time: N/A*	*Software on PC*	Many activities (26 types). However, in the analysis, authors consider only 6 activities. Sleeping, Eating, Personal hygiene, Going out, Relaxing, Others.	Sensing term: 2 months Field: N/A (just a room) Number of participants: 4 **Number of residents: 2**	House A: Algorithm: HMM *Result: Accuracy—61.5%* Cross validation House B: Algorithm: HMM *Result: Accuracy—76.2%* Cross validation.
Placelab [12]	A smart home kit with small/simple state change sensors.	• State change sensors (×77 for Participant 1, and ×84 for Participant 2.) *Installation time: 3 h*	*Software on PDA*	Many activities. However, most of them were not performed. (Participant 1) Preparing lunch, Toileting, Preparing breakfast, Bathing, Dressing, Grooming, Preparing a beverage, Doing Laundry. (Participant 2) Preparing lunch, Listening to music, Toileting, Preparing breakfast, Washing dishes, Watching TV.	*Sensing term: 14 days* Field: N/A (just a room) Number of participants: 2 Number of residents: 1	Participant 1: Algorithm: NB *Result: Accuracy—64.3%* Participant 2: Algorithm: NB *Result: Accuracy—50.8%*
Sweet home [14] /Domus [13]	Multimodal dataset for appliances control using voice in smart home.	• Switch sensor × 8 • Door contact × 6 • PID IR × 2 • Microphone × 7 *Installation time: N/A*	*Software on PC*	Sleeping, Resting Dressing/undressing, Preparing a meal, Having a meal, Doing a laundry, Hygiene activity	Sensing term: 3 h~6 h *Field: Smart home* Number of participants: 11~23 Number of residents: N/A	Algorithm: MLN, SVM, NB Result: *Accuracy with MLN—85.3%* *Accuracy with SVM—59.6%* *Accuracy with NB—66.1%*
CASAS [15]	A smart home kit that can be expanded with minimal effort.	• Motion/Light sensor × 24 • Door sensor × 1 • Relay × 2 • Temperature × 2 and more sensors depending on dataset. **Installation time: Just over 1 h.**	*Software on PC*	Bed-toilet transition, Cook, Eat, Enter home, Leave home, Personal hygiene, Phone, Relax, Sleep, Work.	Sensing term: 1 month *Field: Smart Apart* Number of participants: 18 Number of residents: 1	Algorithm: SVM **Result: F measure—58.9**% 5-fold cross validation.
Our Study	A smart house kit for the elderly in ordinary households.	• Motion sensors × 10 (maximum) • Ambient sensors × 10 (maximum) • Door sensors × 2 (maximum) **Installation time: approximately 45 min**	**Pushbuttons × 5** **(for each activity)**	Bathing, Cooking, Eating, Going out, Sleeping	Sensing term: 2 months **Field: Ordinary Home** Number of participants: 17 **Number of residents: 1 or 2**	Algorithm: LSTM **Result: F measure—40.7%** Holdout validation

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
