# Peer review of "SALON: Simplified Sensing System for Activity of Daily Living in Ordinary Home"

_sensors, 2020, doi:10.3390/s20174895_

Round 1

Reviewer 1 Report

The paper is now ready to be published

Reviewer 2 Report

Overall an interesting paper describing an IoT for health deployment in Japan. The paper is well written and suitable for the journal. Specific comments follow.

  • in the discussion of low cost solutions for the elderly, deployed in the wild, it would be good to reference the SPHERE project, which deployed into 100 homes in the UK [1-3]. Re the requirements: (i) The cost of equipment was <£1,000 per home; (ii) This was not completely solved, but was mitigated by doing Machine Learning training during the installation process; (iii) see below re cameras, otherwise similar to other setups using a combination of passive sensors and wearable devices, and the setup was similar with an in-house server (also a NUC) communicating over LTE with limited access via a custom VPN; (iv) this one was the toughest, with technicians needing to visit every 3-6 months, and installations were tested for a year; (v) there was a very simple companion app that was given to the residents on a tablet computer, enabling them to stop/start/delete recordings, and contact the researchers. Various ML methods were used for both location and activity recognition on the data
  • I don't necessarily agree that just by having cameras, you violate privacy. In the SPHERE project mentioned above, as well as others, great lengths are taken to ensure privacy with video data. One option is to compute some basic features from images (bounding boxes, skeletons, silhouettes) and discard the rest of the video (which has the side benefit of reducing storage requirements and power consumption). User studies have shown this to be acceptable from a privacy perspective, whilst providing a rich source of information
  • Table 1 looks only partially filled - is there really no information for Domus/Sweet Home?
  • The ML approach seems reasonable, and the discussion and methods for dealing with missing data are good to see.
  • The actual accuracies for activity recognition are on the low side -- comparable with CASAS which uses a similar sensor setup. In general, I would argue that at these levels, they are too low to be useful. This will always be a trade-off, but I think the desiderata (i-v) are too strict here (with the exception of iii).

[1] Diethe, Tom, et al. "Releasing eHealth analytics into the wild: lessons learnt from the sphere project." Proceedings of the 24th ACM SIGKDD International Conference on Knowledge Discovery & Data Mining. 2018.

[2] McConville, Ryan, et al. "Vesta: A digital health analytics platform for a smart home in a box." Future Generation Computer Systems (2020).

Reviewer 3 Report

The paper is very interesting and well structured. It would be useful to understand how the sensor system described in your paper can also be used by elderly people but with disabilities (for example, visually impaired).

Author Response

This manuscript is a resubmission of an earlier submission. The following is a list of the peer review reports and author responses from that submission.

Round 1

Reviewer 1 Report

This paper proposed a simplified sensing system for activity of daily living in ordinary home for the elderly. The designed system consists of a small number of inexpensive energy harvesting sensors and simple annotation buttons. Based on the collected binary data, the LSTM is used to recognize the five fixed activities. The designed system was deployed in ten ordinary homes and collected data over two months. This paper detailed the components of the sensing system and gave the activity recognition results.

The issue is of great significance for coping with the aging problem and has been studied for a long time. However, the contribution of this paper is not obvious. The proposed sensing system employs a small number of simple sensors, but the F-measure is very low. Another contribution of this paper is that the proposed system has a simple user interface for the elderly to annotate the activities. However, no matter how simple the annotation tool is, it interferes with the life of the elderly and brings inconvenience to the elderly.

Besides, there are some other problems that authors should be considered.

  1. In page 6, Figure 1 gives the system overview. However, the structure of the system and the icons are confusing. For instance, the circle with blue color denotes BLE, ambient sensor and part of sensor server in the same figure.
  2. In page 6, lines 236-237, the paper introduces the STM431J (https://www.switchscience.com/catalog/2012/) as the energy harvesting module made by Rohm. However, according to the providing website, the STM431J is only a temperature sensor.
  3. In page 14, the design of Exps. (1) and (2) should be explained. Meanwhile, the notation should be introduced.
  4. In page 15, the data showed in Figure 11 is a part of the data listed in Table 4. Therefore, Figure 11 is not necessary.
  5. Please explain the reason why the activity recognition results have low precision.
  6. In Page 15, lines 497-499, authors find that precision and recall are relatively low in multi-resident homes and homes with pets. However, the data listed in Table 4 can not prove this finding. Please provide more data statistics.
  7. Besides, this paper doesn’t compare the activity recognition results with the existing work.
  8. In the references, the format is not unified. For instance, some references use “pp.” while some don’t. The reference [9] is incomplete.

In general, the novelty of this paper is not satisfied. The content can be improved. Therefore, this paper is not suggested to be accepted.

Author Response

Please check the attached response letter.

Reviewer 2 Report

In the proposed manuscript authors are presenting a system for sensing of daily living activities. The system is mainly focused on sensing daily activities of elderly people. Other systems have been proposed in the past but, as authors state, have mostly been tested in experimental environments. In the paper authors present a system, that is inexpensive, robust, maintenance-free, and with a simple interface. The system consists of a small number of energy harvesting sensors and simple annotation buttons. For evaluation purposes, the system was installed in ten typical homes with elderly residents and collected data over a two-month period. Activity recognition was then performed using long short-term memory (LSTM) model. Five typical activities were logged: bathing, cooking, eating, going-out, and sleeping.  A recall rate of 72.3% was achieved on average.

The manuscript is well organized and well written. It reads fluently and only minor errors are detected.

Comments:

In Table 1 it is shown, that the SHiB system fulfills all requirements (suggested by the authors in the manuscript), except the requirement (v). But in the text (p. 5, line 200) authors state, that SHiB kits do not fulfill requirements (iv) and (v). This should be clarified.

Page 10, line 384: …ADL data and would thus useful for anomaly detection and/or activity recognition. I think there is a “be” missing.

Author Response

(The authors gave the same response as above.)

Reviewer 3 Report

This paper presents an ADL sensing system that takes into account requirements such as inexpensiveness, robustness, privacy, maintenance-free, and user-friendliness.

The paper is well written and organized, and it is my opinion that it could be of interest for the reader of this journal.

However, there are some issue that should be addressed before considering it for publication:

  • In their analysis, the authors did not consider one important dimension that is the possibility of cognitive/physical impairments for the elderly. This, of course, can pose special requirements to be taken into account and may require different techniques to be adopted for activity recognition.
    In particular, authors adopted pattern matching techniques, but there are others that may be needed in order to face the peculiarities of cognitively impaired people.
    Authors should at least discuss the problem presented in:
    1. “Cognitive Assisted Living Ambient System: A Survey”. Published in Digital Communications and Networks;
  • Another aspect that has not been faced is the situation awareness dimension. Authors focused on context awareness, but activity recognition can effectively improve if conducted at a higher level, such as the situational-awareness one.
    In addition to some of the previous point, I suggest to consider the following work:
    1. “Situation identification techniques in pervasive computing: A review”, Pervasive and Mobile Computing

  • Finally, in addition to security and privacy, another relevant dimension that should be considered is dependability/reliability. For this reason, authors could refer to works like:
    1. “Dependability of wireless sensor networks for industrial prognostics and health management”, Computers in industry
    2. “A formal methodology to design and deploy dependable wireless sensor networks”, Sensors
    3. “A Survey on Data Quality for Dependable Monitoring in Wireless Sensor Networks”, Sensors
    4. “A comprehensive wireless sensor network reliability metric for critical Internet of Things applications”, Journal on Wireless Communications and Networking
    5. “Wireless Sensing Networks System Dependability Measurement”, Journal of sustainable development

Author Response

(The authors gave the same response as above.)

Round 2

Reviewer 1 Report

The paper lacks novelty and contribution. No comparison with the related work.

Reviewer 3 Report

This new version of the paper presents the following paragraph in the final section:

"Since the proposed system handles highly privacy-sensitive data, it is necessary to consider security issues. Therefore, in order to protect the data from external threats, it is important to make the wireless sensor network more secure [60–64], and to encourage residents to make appropriate decisions about which sensor to use [65]."

The authors mix under the "security" umbrella" papers ([60-64] and [65]) most of which are not related to security.

Differently, such articles are related to dependability, reliability, formal specification (of reliability characteristics) and quality of data.
To the best of my knowledge, such topics are different quality attributes from security.